# TIM-3 Expression on Dendritic Cells in Colorectal Cancer

**DOI:** 10.3390/cancers16101888

**Published:** 2024-05-15

**Authors:** Mei Sakuma, Masanori Katagata, Hirokazu Okayama, Shotaro Nakajima, Katsuharu Saito, Takahiro Sato, Satoshi Fukai, Hideaki Tsumuraya, Hisashi Onozawa, Wataru Sakamoto, Motonobu Saito, Zenichiro Saze, Tomoyuki Momma, Kosaku Mimura, Koji Kono

**Affiliations:** 1Department of Gastrointestinal Tract Surgery, Fukushima Medical University, Fukushima 960-1295, Japan; msakuma@fmu.ac.jp (M.S.); okayama@fmu.ac.jp (H.O.);; 2Department of Multidisciplinary Treatment of Cancer and Regional Medical Support, Fukushima Medical University, Fukushima 960-1295, Japan; 3Department of Blood Transfusion and Transplantation Immunology, Fukushima Medical University, Fukushima 960-1295, Japan

**Keywords:** TIM-3, dendritic cell, colorectal cancer, cGAS-STING, immunotherapy

## Abstract

**Simple Summary:**

Since TIM-3 on T cells or dendritic cells (DCs) has recently emerged as an attractive target for immunotherapy, we examined its expression on DCs using transcriptomic data from a public database and immunohistochemical evaluations from our cohorts of colorectal cancer. The expression of *HAVCR2* (TIM-3) was strongly associated with the infiltration of DCs within the tumor microenvironment, and immunohistochemical staining of clinical tissue samples revealed that tumor-infiltrating DCs expressed TIM-3; however, their number at the tumor-invasive front significantly decreased with stage progression. Among vitro-generated DCs, TIM-3 expression was higher on immature DCs than on mature DCs, while Western blotting showed that STING expression was higher on mature DCs than on immature DCs.

**Abstract:**

TIM-3 was originally identified as a negative regulator of helper T cells and is expressed on dendritic cells (DCs). Since the inhibition of TIM-3 on DCs has been suggested to enhance T cell-mediated anti-tumor immunity, we examined its expression on DCs within the tumor microenvironment (TME) in colorectal cancer (CRC) using transcriptomic data from a public database (*n* = 592) and immunohistochemical evaluations from our cohorts of CRC (*n* = 115). The expression of TIM-3 on DCs in vitro was examined by flow cytometry, while the expression of its related molecules, cGAS and STING, on immature and mature DCs was assessed by Western blotting. The expression of *HAVCR2* (TIM-3) was strongly associated with the infiltration of DCs within the TME of CRC. Immunohistochemical staining of clinical tissue samples revealed that tumor-infiltrating DCs expressed TIM-3; however, their number at the tumor-invasive front significantly decreased with stage progression. TIM-3 expression was higher on immature DCs than on mature DCs from several different donors (*n* = 6). Western blot analyses showed that the expression of STING was higher on mature DCs than on immature DCs, which was opposite to that of TIM-3. We demonstrated that TIM-3 was highly expressed on tumor-infiltrating DCs of CRC and that its expression was higher on immature DCs than on mature DCs.

## 1. Introduction

Colorectal cancer (CRC) is the third leading cause of cancer-related death and the second most frequent cancer worldwide (GLOBOCAN 2020) [1]. Although patients with advanced CRC receive multidisciplinary treatments, including surgical resection combined with chemotherapy and radiotherapy, they still have a poor prognosis, and, thus, the development of new strategies is required.

Immune checkpoint inhibitors (ICIs), such as antibodies against programmed cell death-1 (PD-1) and its ligand programmed death ligand 1 (PD-L1) and cytotoxic T-lymphocyte-associated protein 4 (CTLA-4), have emerged as a new effective therapy for several solid cancers [2]. Although the findings of initial trials on ICIs for CRC were not promising, the mismatch-repair deficiency (dMMR)/microsatellite instability-high (MSI-H) subset is currently regarded as an excellent target for anti-PD-1/anti-PD-L1 immunotherapy [3,4]. MSI-H tumors appear to be immunogenic; dMMR/MSI-H induces hypermutations and tumor neoantigens, contributing to the induction of the “immune hot” tumor microenvironment. In several clinical trials, anti-PD-1 antibodies, with or without the blockade of CTLA-4, achieved marked and durable responses with prolonged survival for metastatic MSI-H CRC patients [3,4]. However, since approximately 15% of CRC patients have MSI-H tumors [5], ICIs are only available for a small percentage of CRC patients.

T cell immunoglobulin and mucin domain (TIM)-3 was originally identified as a negative regulator of helper T cells [6]. TIM-3, encoded by the *HAVCR2* gene, has recently become an attractive target for immunotherapy because it is frequently co-expressed with PD-1 on exhausted T cells [7,8]. TIM-3 may be involved in the mechanisms underlying the acquisition of resistance to anti-PD-1 therapy because its expression was found to be up-regulated in response to anti-PD-1 therapy. Therefore, the co-blockade of TIM-3 and PD-1 may overcome resistance to anti-PD-1 therapy [8,9]. Moreover, TIM-3 is expressed on myeloid cells, such as monocytes, macrophages, and dendritic cells (DCs), and the expression of TIM-3 was found to be higher on DCs in tumor microenvironments than on those in normal tissues [6]. Since several clinically available monoclonal antibodies (mAbs) against TIM-3 have been evaluated in clinical trials [10,11], targeting TIM-3 on DCs, macrophages, and exhausted T cells with/without anti-PD-1 therapy has potential as a novel treatment strategy.

DCs are professional antigen-presenting cells that take up tumor antigens and educate anti-tumor-specific T cells in tumor-bearing hosts. DCs are classified as conventional DCs (cDCs) consisting of two subsets (cDC1 and cDC2), plasmacytoid DCs (pDCs), inflammatory DCs, and Langerhans cells [12]. cDC1 is characterized by the expression of the unique C-type lectin receptor Clec9a and the chemokine receptor XCR1. cDC2 is identified by the high expression of MHC II, CD11c, CD1c, and SIRPA. pDC subsets are the third subset of DCs and are characterized by their unique ability to produce large amounts of type I interferon and their consequent role in antiviral immunity [12]. Recent studies demonstrated that the blockade of TIM-3 up-regulated the expression of chemokines by cDC1s, thereby driving T cell effector functions [13], and also that TIM-3 inhibited the activation of the cyclic GMP-AMP synthase (cGAS) stimulator of interferon genes (STING) pathway in intra-tumoral DCs by suppressing the extracellular uptake of DNA [14]. Therefore, the inhibition of TIM-3 on DCs may enhance T cell-mediated anti-tumor immunity, which prompted us to examine the expression of TIM-3 on DCs within the tumor microenvironment.

The cGAS is the direct cytosolic DNA sensor and innate immune response initiator, which binds to cytosolic DNA and activates STING as a downstream adaptor through the generation of the second messenger cyclic GMP-AMP (cGAMP) on tumor cells and DCs [14]. As for the regulatory mechanism for TIM-3, it has been reported that TIM-3 on M2-like macrophages was, at least in part, induced by transforming growth factor-β (TGF-β) in hepatocellular carcinoma, and a SMAD-binding site in the 5′ region of TIM-3 was identified [15]. Therefore, it is possible that the cGAS-STING pathway in DCs might be regulated by TGF-β within the tumor microenvironment.

In the present study, we investigated the expression of TIM-3 using transcriptomic data from a public database and immunohistochemical (IHC) evaluations from our cohorts of CRC and also examined the expression of TIM-3 and its related molecules, cGAS and STING, on immature and mature DCs in vitro.

## 2. Materials and Methods

### 2.1. The Cancer Genome Atlas (TCGA) Dataset Analysis

The mRNA expression z-scores of genes (RNA-Seq V2 RSEM normalized, RNA-Seq data) were downloaded from the TCGA colorectal adenocarcinoma (PanCancer Atlas) dataset (*n* = 592) through cBioPortal (https://www.cbioportal.org/, accessed on 1 October 2022) [16]. In the present study, we analyzed the mRNA expression levels of TIM-3 (*HAVCR2*), DC-SIGN (*CD209*), and CD83 (*CD83*). We utilized multi-gene expression signatures, including cDC1 (*CLEC9A*, *XCR1*, *BATF3*, and *CLNK*), cDC2 (*CD1A*, *CD1B*, *FCER1A*, *CD1E*, and *CLEC10A*), and pDC (*IL3RA*, *TCR9*, and *LILRA4*) [17,18,19].

### 2.2. Patient Samples

Colorectal adenocarcinoma patients with stage I-IV who underwent surgical resection at the Gastrointestinal Tract Surgery, Fukushima Medical University Hospital, between April 2019 and November 2021 were included in the present study. Patients with stage III-IV who were treated with preoperative treatment, such as self-expanding metal stents, chemotherapy, or radiotherapy, were also included. Patients who had no tumor in formalin-fixed, paraffin-embedded (FFPE) tumor blocks from surgical specimens were excluded. Pathological stages were diagnosed according to the Japanese Classification of Colorectal, Appendiceal, and Anal Carcinoma (9th Edition). Patient characteristics are shown in Table 1. The present study was approved by the Ethical Committee of Fukushima Medical University (Reference No. 2289), and all patients provided written informed consent before enrollment.

### 2.3. IHC

FFPE sections (thickness of 4 μm) were deparaffinized in xylene and rehydrated in ethanol. Endogenous peroxidases were blocked with 0.3% hydrogen peroxide in methanol. Double staining was performed using the SignalStain IHC Dual Staining Kit (#36084, Cell Signaling Technology, Danvers, MA, USA). Slides were incubated with a primary mouse monoclonal anti-LAMP-3 antibody (clone 16H11.2, #3503452, Merck, Darmstadt, Germany; 1:500) at 4 °C overnight. Sections were then incubated at room temperature for 30 min with a secondary horseradish peroxidase (HRP)-conjugated mouse antibody (SignalStain Boost IHC Detection Reagent, Cell Signaling Technology). Visualization was performed with diaminobenzidine (SignalStain DAB Substrate Kit, Cell Signaling Technology). Antigens were retrieved by autoclaving for 10 min in Target Retrieval Solution (Agilent Technologies, Santa Clara, CA, USA) (100 °C, pH 9.0) for TIM-3. Slides were then incubated with a primary rabbit monoclonal anti-TIM-3 antibody (D5D5R XP, #45208, Cell Signaling Technology; 1:200) at 4 °C overnight. Slides were treated with SignalStain Boost IHC Detection Reagent (AP, Rabbit) (Cell Signaling Technology), and visualization was performed with Vibrant Red (SignalStain Vibrant Red Alkaline Phosphatase Substrate Kit, Cell Signaling Technology). Counterstaining was conducted with hematoxylin.

### 2.4. IHC Assessment

Digital images were acquired with a digital slide scanner (NanoZoomer-SQ; Hamamatsu Photonics, Shizuoka, Japan). Scanned images were independently evaluated by two observers (M. Sakuma and S. Nakajima) who had no prior knowledge of any clinical data. CD208-positive (+) cells were identified by brown staining and TIM-3 (+) cells by red staining. Four independent areas in each tumor center and the invasive front region of each tumor were randomly reviewed and evaluated by counting the number of CD208 (+) cells and both CD208 and TIM-3 (+) cells at a magnification of ×200.

### 2.5. Generation of DCs

The generation of DCs was performed using the ImmunoCult^TM^ Dendritic Cell Culture Kit (10985, STEMCELL Technologies, Vancouver, BC, Canada) according to the manufacturer’s protocol. Human monocytes were isolated by RosetteSep™ Human Monocyte Enrichment Cocktail (#15068, STEMCELL Technologies) from the peripheral blood samples of healthy volunteers collected using BD Vacutainer^®^ CPT™ (Becton Dickinson and Company, Franklin Lakes, NJ, USA). Monocytes were cultured in a 12-well plate at a density of 1 × 10^6^ per well at 37 °C with a 5% CO_2_ atmosphere in ImmunoCult^TM^-ACF Dendritic Cell Medium (STEMCELL Technologies) supplemented with ImmunoCult^TM^-ACF Dendritic Cell Differentiation Supplement (STEMCELL Technologies) for 3 days. After 3 days, the medium was replaced with fresh medium containing a differentiation supplement, and cells were cultured for 2 days. After 2 days, the harvested cells were used as immature DCs. To generate mature DCs, ImmunoCult^TM^-ACF Dendritic Cell Maturation Supplement (STEMCELL Technologies) was added to the same medium, and cells were cultured for an additional 2 days.

### 2.6. TGF-β Treatment and Anti-TIM-3 Antibody Treatment

After generating immature DCs and mature DCs from peripheral blood mononuclear cells, each cell was exposed to 0, 1, 10, or 100 ng/mL of TGF-β1 (#7754-BH, R&D Systems, Minneapolis, MN, USA). Cells were harvested 24, 48, or 72 h later and subjected to flow cytometry. Regarding Western blotting, immature DCs and mature DCs were exposed to 100 ng/mL of TGF-β1 and 10 μg/mL of the Ultra-LEAF^TM^ anti-human TIM-3 antibody (#345009, BioLegend, San Diego, CA, USA) at the same time. After a 48-h incubation, cells were collected.

### 2.7. Flow Cytometry

The immature DCs and mature DCs collected were subjected to flow cytometry, and staining was performed according to the manufacturer’s protocol for each antibody. The following antibodies were used in the present study: APC/Cyanine7 mouse anti-human CD14 mAb (#325620, BioLegend), FITC mouse anti-human CD209 (#330103, BioLegend), Pacific Blue mouse anti-human CD86 mAb (#305423, BioLegend), and PE mouse anti-human CD366 (TIM-3) mAb (#345005, BioLegend). An unstained sample was used as a negative control, and dead cells were detected using 7AAD (#420404, BioLegend). Stained cells were measured using the BD FACS Canto II flow cytometer (BD Biosciences, Franklin Lakes, NJ, USA), and data were analyzed with FlowJo software (version 10.8.1, FlowJo, Ashland, OR, USA).

### 2.8. Gating Methods

We used forward scatters and side scatters to gate the population of immune cells, including DCs, followed by the classification of single cells and then 7AAD-negative cells. In the analysis of TIM-3 expression on immature DCs and mature DCs, CD209 (+) cells were examined as immature DCs and CD14 (−) CD86 (+) cells as mature DCs.

### 2.9. Western Blot Analysis

Total cell proteins were extracted using RIPA buffer supplemented with a protease inhibitor and phosphatase inhibitor cocktail. Protein concentrations were measured using the XL-Bradford kit (Aproscience, Tokushima, Japan). Tris-glycine SDS sample buffer was added to protein samples and boiled at 100 °C for 5 min. Ten micrograms of protein were loaded onto a 4–20% Tris-Glycine gel (Thermo Fisher Scientific, Waltham, MA, USA) and electrophoresed at 125 V for 60 min using the Invitrogen XCell SureLock electrophoresis system with Tris-Glycine SDS Running Buffer (Thermo Fisher Scientific). Proteins were then transferred onto PVDF membranes using the iBlot2 Dry Blotting System (Thermo Fisher Scientific). Membranes were blocked with 5% non-fat dried skimmed milk and incubated with primary antibodies, including anti-STING mAb (#13647, Cell Signaling Technology), anti-cGAS (#79978, Cell Signaling Technology), and anti-β-actin mAb (sc-69879, Santa Cruz Biotechnology, Dallas, TX, USA) at 4 °C overnight. The membranes were then incubated with HRP-linked anti-mouse IgG or anti-rabbit IgG antibodies (Cell Signaling Technology) at room temperature for 1 h. Immunoreactive proteins were visualized using ImageQuant LAS 4000 mini (Fuji Film, Tokyo, Japan) with ECL prime Western blot detection reagent (GE Healthcare, Chicago, IL, USA).

### 2.10. Statistical Analysis

Statistical analyses were performed using Graph Pad Prism 9 (Graph Pad Software, San Diego, CA, USA). Two groups were compared using the Student’s *t*-test, and multiple groups were compared using the Kruskal–Wallis test with Dunn’s post hoc test. Correlations were examined using Spearman’s coefficient. All error bars indicate the mean ± standard deviation, and a value of *p* < 0.05 was considered to be significant.

## 3. Results

### 3.1. Correlation between the Expression of HAVCR2 and DC Subpopulations in CRC

We investigated whether the infiltration of DCs was associated with *HAVCR2* expression (which encodes TIM-3) in CRC using the TCGA dataset. The results obtained revealed correlations between *HAVCR2* expression levels and a panel of DC signatures (Figure 1). For example, *HAVCR2* correlated with the cDC1 signature consisting of *CLEC9A*, *XCR1*, *BATF3*, and *CLNK* [17], with the cDC2 signature consisting of *CD1A*, *CLEC1B*, *FCER1A*, *CD1E*, and *CLEC10A* [18,19], and also with the pDC signature consisting of *IL3RA*, *TCR9*, and *LILRA4* [18,19]. Furthermore, correlations were observed between *HAVCR2* and *CD209* (an immature DC marker) and between *HAVCR2* and *CD83* (a mature DC marker). 

Collectively, these results indicated that the expression of *HAVCR2* correlated with the infiltration of DCs within the tumor microenvironment of CRC.

### 3.2. TIM3 Expression on DCs of CRC by IHC

Since a reliable IHC marker for mature DCs is CD208 [20,21], we examined the expression of CD208 in relation to that of TIM-3 within the tumor microenvironment of CRC using a double staining technique with IHC. As shown in Figure 2, it was possible to specifically identify CD208 (+) cells and TIM-3 (+) cells at both the tumor center and invasive front, indicating that tumor-infiltrating DCs expressed TIM-3 at the protein level. The number of CD208 (+) cells (mature DCs) at the tumor invasive front significantly decreased with stage progression (Figure 3). Moreover, the prevalence of TIM-3 (+) on mature DCs (TIM-3 (+)/CD208 (+)) at the tumor-invasive front of CRC also decreased with stage progression.

Collectively, these results indicate that tumor-infiltrating DCs expressed TIM-3; however, their numbers at the invasive front significantly decreased with stage progression.

### 3.3. TIM-3 Expression Was Higher on Immature DCs Than on Mature DCs

We generated immature and mature DCs from peripheral blood monocytes using the in vitro induction method [22,23]. The phenotypes of immature and mature DCs were qualitatively confirmed by flow cytometry (Appendix A). TIM-3 expression was higher on immature DCs than on mature DCs, which was confirmed using several different donors (*n* = 6, Figure 4A). Since a previous study reported that TGF-β up-regulated the expression of TIM-3 on monocytes or M2 macrophages [15], we examined the expression of TIM-3 in response to TGF-β at several different doses and intervals. The results obtained showed that TGF-β did not affect the expression of TIM-3 on DCs (Figure 4A and Appendix A).

We then investigated cGAS and STING expression in immature and mature DCs by Western blotting based on previous findings showing that TIM-3 expression was negatively associated with the cGAS and STING pathways in DCs [14]. As shown in Figure 4B, the expression of STING was higher on mature DCs than on immature DCs, which was opposite to that for TIM-3 expression, while the expression of cGAS remained unchanged. Moreover, the expression of cGAS and STING was not affected by TGF-β or anti-TIM-3 mAb without ligand stimulation (Figure 4B). The original images of the Western Blotting figures can be found in Appendix A.

## 4. Discussion

In the present study, we demonstrated that TIM-3 was highly expressed on DCs within tumor microenvironments, at both the tumor center and invasive front of CRC, which is supported by transcriptional data from a public database and an IHC analysis of our own cohorts. Moreover, the expression of TIM-3 was higher on immature DCs than on mature DCs, along with the down-regulation of cGAS-STING, according to the in vitro system.

TIM-3, encoded by the *HAVCR2* gene, has recently emerged as an immunotherapy target because it is co-expressed with PD-1 on dysfunctional and exhausted tumor-infiltrating T cells. Moreover, TIM-3 was shown to be highly expressed on several types of myeloid cells, including DCs, and myeloid cells have been suggested to play a predominant role in TIM-3-mediated anti-tumor immunity [6,24,25]. Dixon et al. reported, using the conditional knockout of TIM-3 together with single-cell RNA sequencing, the singular importance of TIM-3 on DCs, whereby the loss of TIM-3 on DCs promotes strong anti-tumor immunity [26]. Therefore, targeting TIM-3 on DCs represents an attractive and promising approach to enhancing anti-tumor immunity. However, the expression profile of TIM-3 and the underlying mechanisms by which it is regulated on DCs of the CRC remain unknown. In the present study, we showed that DCs highly expressing TIM-3 were present at both the tumor center and invasive front of CRC, suggesting that TIM-3 on DCs is targetable within the tumor microenvironment. However, we need to consider decreases in the prevalence of DCs expressing TIM-3 at the invasive front with stage progression. Therefore, early-stage CRC may be a good target population for immunotherapy targeting TIM-3 in DCs.

Regarding the regulatory mechanism for TIM-3, a previous study reported that TIM-3 on M2-like macrophages was, at least in part, induced by TGF-β in hepatocellular carcinoma, and a SMAD-binding site in the 5′ region of TIM-3 was identified [15]. However, in the present study, the regulation of TIM-3 expression on DCs was not associated with TGF-β. Since TIM-3 expression on T cells was found to be up-regulated by IL-2 [27], the regulatory mechanisms for TIM-3 may differ between cell types, and thus, further study is needed on the regulatory mechanisms of TIM-3 on DCs.

Regarding downstream elements of TIM-3 on DCs, TIM-3 may prevent the cytoplasmic localization and activation of cGAS-STING on DCs, where the cGAS-STING pathway is fundamental for exerting anti-tumor immunity [14]. In the present study, we confirmed in the in vitro model that the expression of TIM-3 was higher on immature DCs than on mature DCs, while STING levels were conversely lower on immature DCs than on mature DCs. Therefore, TIM-3 appears to play an important role in the negative control of downstream elements, such as cGAS-STING on DC.

TIM-3 has potential as an actionable target in combination with PD-1/PD-L1 blockade for CRC patients because the dual inhibition of PD-1 and TIM-3 exerted synergic effects in animal models [8,28]. Furthermore, acquired resistance to PD-1 blockade was associated with the up-regulated expression of TIM-3 [9,29]; therefore, targeting TIM-3 may be effective even in cases where patients do not respond to anti-PD-1/PD-L1 immunotherapy. Moreover, among TIM-3-expressing cells, including T cells, NK cells, and DCs, DCs were the most potent regulators of anti-tumor immunity by the TIM-3 deletion system specific to cell types in mouse models [26]. The expression of TIM-3 may also perform as a marker of a poor prognosis in some cancer types, including CRC [30,31,32]. Therefore, anti-TIM-3 antibodies, mainly in combination with anti-PD-1/PD-L1 immune checkpoint inhibition, are currently being evaluated in clinical trials for the treatment of advanced cancers, including CRC [10,33]. Therefore, in the present study, our observations for TIM-3 expression on DCs further support the possibility and rationale for anti-TIM-3 immunotherapy in CRC.

There are several limitations to the present work. First, the number of study cohorts was small. Second, although we have shown in the present study that TIM-3 expression was more dominant on immature DCs than on mature DCs, along with the down-regulation of cGAS-STING, more precise clarification of the regulatory mechanism for TIM-3 on DCs would be needed in future studies.

## 5. Conclusions

A transcriptomic data analysis, IHC, and in vitro experiments showed that TIM-3 was highly expressed on tumor-infiltrating DCs in CRC, and its expression was higher on immature DCs than on mature DCs.

## Figures and Tables

**Figure 1 cancers-16-01888-f001:**
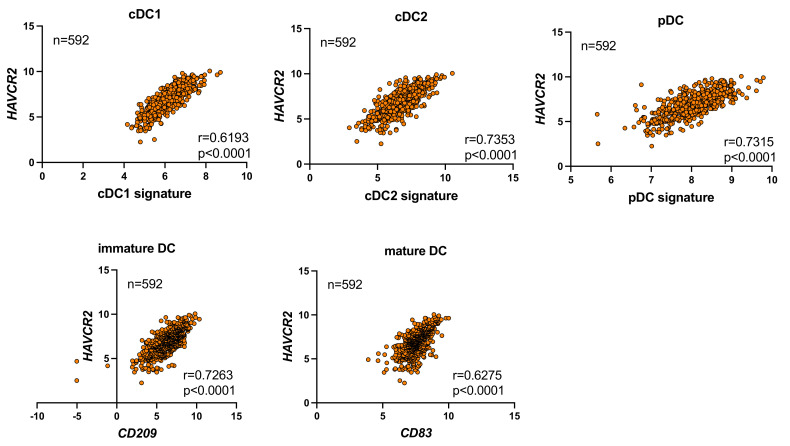
Correlations between the expression of *HAVCR2* and DC subpopulations in CRC. We utilized multi-gene expression signatures, including cDC1 (*CLEC9A*, *XCR1*, *BATF3*, and *CLNK*), cDC2 (*CD1A*, *CD1B*, *FCER1A*, *CD1E*, and *CLEC10A*), and pDC (*IL3RA*, *TCR9*, and *LILRA4*). We evaluated the mRNA expression levels of *CD209* (an immature DC marker) and *CD83* (a mature DC marker). Spearman’s coefficients are indicated (*p* < 0.0001).

**Figure 2 cancers-16-01888-f002:**
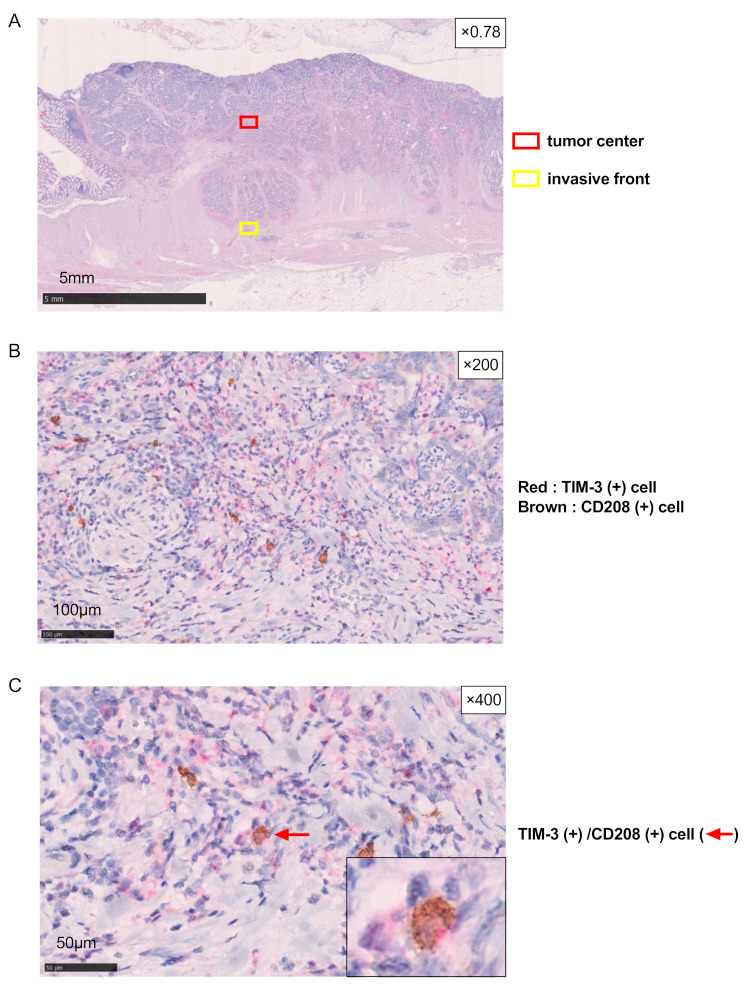
Immunohistochemistry for mature DCs and TIM-3 (+) cells in CRC tissues. (**A**) Four independent areas at each tumor center and the invasive front region of each tumor were randomly reviewed. (**B**,**C**) A representative case of CRC showing CD208 (+) and/or TIM-3 (+) cells (**B**,**C**) and TIM-3 (+)/CD208 (+) cells (**C**).

**Figure 3 cancers-16-01888-f003:**
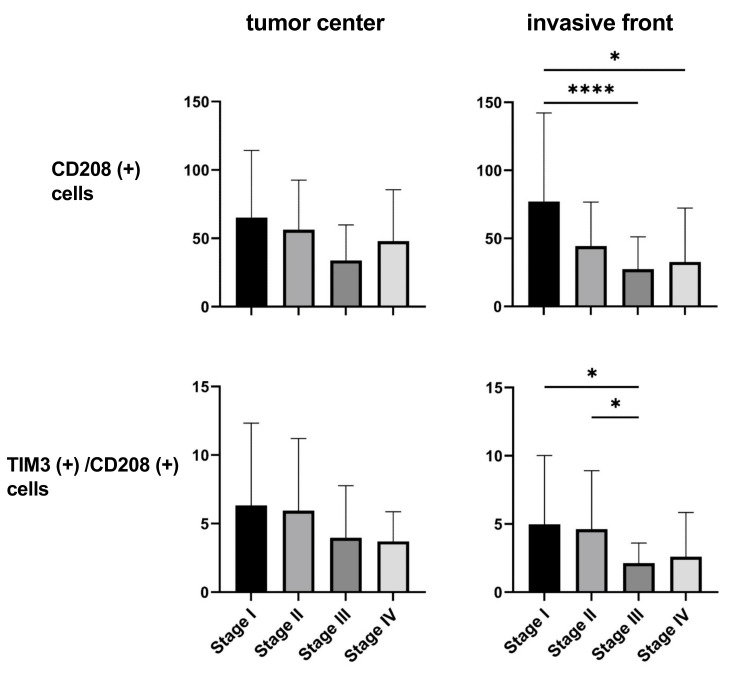
Numbers of CD208 (+) cells and TIM-3 (+)/CD208 (+) cells at both the tumor center and invasive front (*n* = 115). The Kruskal–Wallis test with Dunn’s post hoc test. * *p* < 0.05, **** *p* < 0.0001.

**Figure 4 cancers-16-01888-f004:**
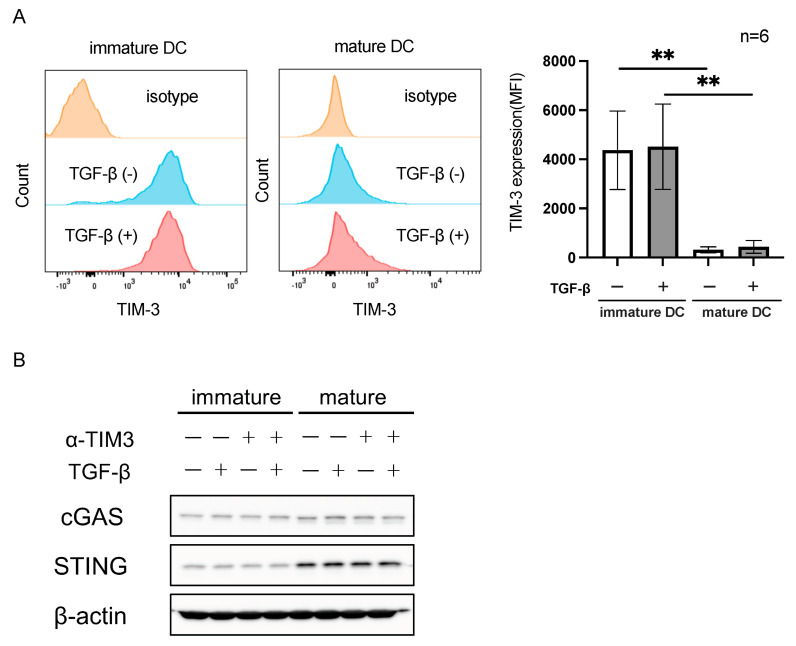
TIM-3 expression on DCs and TGF-β-treated DCs. (**A**) Representative histograms for immature and mature DCs are shown, and the expression levels of TIM-3 on immature and mature DCs were compared using a paired *t*-test (*n* = 6). ** *p* < 0.01. (**B**) Expression of cGAS and STING on immature and mature DCs by Western blotting, treated with TGF-β and anti-TIM-3 mAb (α-TIM3) without ligand stimulation as indicated.

**Table 1 cancers-16-01888-t001:** Clinicopathological characteristics of 115 patients with CRC.

Characteristics		N (% or Median Range)
Sex		
	Male	69 (60.0)
	Female	46 (40.0)
Age (years)		71 (30–88)
Location		
	Proximal colon	32 (27.8)
	Distal colon	32 (27.8)
	Rectum	51 (27.8)
Differentiation		
	Well	46 (40.0)
	Moderately	56 (48.7)
	Poorly	2 (1.7)
	Mucinous	7 (6.1)
	Other	4 (3.5)
Tumor invasion		
	T1	20 (17.4)
	T2	19 (16.5)
	T3	43 (37.4)
	T4	33 (28.7)
Lymph node metastasis		
	Absent	76 (66.1)
	Present	39 (33.9)
Distant metastasis		
	Absent	105 (91.3)
	Present	10 (8.7)
Stage		
	I	34 (29.6)
	II	39 (33.9)
	III	32 (27.8)
	IV	10 (8.7)
Preoperative treatment		
	No treatment	97 (84.3)
	Chemotherapy only	9 (7.8)
	Self-expanding metal stent	7 (6.1)
	Chemotherapy + self-expanding metal stent	1 (0.9)
	Radiotherapy	1 (0.9)

## Data Availability

The datasets generated and/or analyzed during the present study are available from the corresponding author upon reasonable request.

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
