# Peer review of "TIM-3 Expression on Dendritic Cells in Colorectal Cancer"

_cancers, 2024, doi:10.3390/cancers16101888_

Round 1

Reviewer 1 Report

Comments and Suggestions for Authors

In this article “TIM-3 expression on dendritic cells in colorectal cancer”. The authors have used transcriptomic data from a public database and their clinical immunohistochemical evaluations to assess TIM-3 expression and its corresponding molecules in immature and in mature dendritic cells in vitro. Nevertheless, there are some recommendations to further improve the article:

1-       It is recommended to define all abbreviations.

2-       In the introduction section, it is highly recommended to extend this part by introducing a paragraph concerning transforming growth factor (TGF-β) and cGAS–STING signaling pathway.

3-       In Figure 3 A, the asterisk, which indicates significant differences, is missing above the graph bars.

4-       I highly recommend that the authors add the title of the manuscript and the name of the authors in the supplementary materials.

5-       On page 9, line 265, the author’s name, Dixon, is hyperlinked to the National Library of Medicine. Please remove this hyperlink.

6-       It is highly recommended to include in the section of discussion the limitations of this work and how you can improve it in your future works.

7-       The authors should revise the manuscript extensively and correct all grammatical errors.

8-       There are some grammatical suggestions to improve the manuscript:

·        p. 1, line 17: The term "number" suggests that there must be "the number".

·        p. 1, line 19: The term "on immature" suggests that there must be "in immature".

·        p. 1, line 19: The term "than mature" suggests that there must be "than in mature".

·        p. 1, line 23: The term "expression" suggests that there must be "the expression".

·        p. 2, line 50: The term "tumors seems" suggests that there must be "tumors seem".

·        p. 2, line 51: The term " to induction" suggests that there must be "to the induction".

·        p. 2, line 60: The term " of importance" suggests that there must be "It is of importance".

·        p. 2, line 61: The term " in acquired" suggests that there must be "in the acquired".

·        p. 2, line 68: The term "as well as" suggests that there must be a "and".

·        p. 2, line 74: The term "of unique" suggests that there must be "of the unique".

·        p. 2, line 77: The term " the consequent" suggests that there must be "their consequent".

·        p. 3, line 101: The term "excluded in" suggests that there must be "excluded from ".

·        p. 3, line 103: The term " were shown" suggests that there must be "are shown".

·        p. 3, line 113: The term "with secondary" suggests that there must be "with a secondary".

·        p. 3, line 133: The term " was used" suggests that there must be "was performed using".

·        p. 4, line 145: The term " for additional" suggests that there must be "for an additional".

·        p. 5, line 165: The term "by classification" suggests that there must be "by the classification".

·        p. 6, line 212: The term "that reliable" suggests that there must be a " that a reliable".

·        p. 6, line 217: The term " Then, number" suggests that there must be a " Then, the number".

·        p. 7, line 226: The term "each the tumor" suggests that there must be "each tumor".

·        p. 8, line 233: The term "on immature" suggests that there must be a "in immature".

·        p. 8, line 233: The term "than mature" suggests that there must be a "than in mature".

·        p. 8, line 237: The term "by confirming with" suggests that there must be a "as confirmed by".

·        p. 8, line 239: The term "macrophage" suggests that there must be "macrophages".

·        p. 8, line 242: The term " we have evaluated" suggests that there must be "we evaluated".

·        p. 8, line 244: The term "pathway" suggests that there must be a " pathways".

·        p. 9, line 260: The term "by evaluating " suggests that there must be "according to".

·        p. 9, line 263: The term "it has been also" suggests that there must be a "it has also been ".

·        p. 9, line 269: The term "to enhance" suggests that there must be "to enhancing".

·        p. 10, line 270: The term "and underlying" suggests that there must be a " and the underlying ".

·        p. 10, line 274: The term "attention the fact" suggests that there must be a "attention to the fact".

·        p. 10, line 274: The term "at invasive front was" suggests that there must be a "at the invasive front has".

·        p. 10, line 278: The term "were" suggests that there must be a "was".

·        p. 10, line 286: The term "to exert" suggests that there must be a "to exerting".

·        p. 10, line 296: The term "who failed to" suggests that there must be a "where patients failed".

·        p. 10, line 298: The term "for anti-tumor" suggests that there must be a "of anti-tumor".

·        p. 10, line 299: The term "for cell types" suggests that there must be a "to cell types".

Comments on the Quality of English Language

In the manuscript entitled “TIM-3 expression on dendritic cells in colorectal cancer”. The authors have written the article in comprehensive English but with extensive grammatical errors.

Reviewer 2 Report

Comments and Suggestions for Authors

Even it is an interesting topic, lacks novelty It is well know the contribution of TIM3 to CRC and immunotherapy

1. the number of patients is small

2. more inclusion/exclusion criteria should be described

3, control population is missing

4. What are the characteristics of patients that used for dendritic cells characterization

5. correlation of the findings with disease stage etc is missing

Round 2

Reviewer 2 Report

Comments and Suggestions for Authors

Still the novelty is missing however the manuscript has been improved